# Feasibility of Non-Gaussian Diffusion Metrics in Chronic Disorders of Consciousness

**DOI:** 10.3390/brainsci9050123

**Published:** 2019-05-27

**Authors:** Elena I. Kremneva, Liudmila A. Legostaeva, Sofya N. Morozova, Dmitry V. Sergeev, Dmitry O. Sinitsyn, Elizaveta G. Iazeva, Aleksandr S. Suslin, Natalia A. Suponeva, Marina V. Krotenkova, Michael A. Piradov, Ivan I. Maximov

**Affiliations:** 1Research Center of Neurology, 80 Volokolamskoe shosse, 125367 Moscow, Russia; milalegostaeva@gmail.com (L.A.L.); kulikovasn@gmail.com (S.N.M.); dmsergeev@yandex.ru (D.V.S.); d_sinitsyn@mail.ru (D.O.S.); lizaveta.mochalova@gmail.com (E.G.I.); razor1911@mail.ru (A.S.S.); nasu2709@mail.ru (N.A.S.); krotenkova_mrt@mail.ru (M.V.K.); mpi711@gmail.com (M.A.P.); 2Department of Psychology, University of Oslo, Forskningsveien 3A, 0373 Oslo, Norway; 3Norwegian Centre for Mental Disorders Research (NORMENT), Norway and Institute of Clinical Medicine, University of Oslo, Oslo Universitetssykehus Bygg 48 Ullevål, 0317 Oslo, Norway

**Keywords:** diffusion MRI, DKI, chronic disorders of consciousness, unresponsive wakefulness syndrome, vegetative state, minimally conscious state

## Abstract

Diagnostic accuracy of different chronic disorders of consciousness (DOC) can be affected by the false negative errors in up to 40% cases. In the present study, we aimed to investigate the feasibility of a non-Gaussian diffusion approach in chronic DOC and to estimate a sensitivity of diffusion kurtosis imaging (DKI) metrics for the differentiation of vegetative state/unresponsive wakefulness syndrome (VS/UWS) and minimally conscious state (MCS) from a healthy brain state. We acquired diffusion MRI data from 18 patients in chronic DOC (11 VS/UWS, 7 MCS) and 14 healthy controls. A quantitative comparison of the diffusion metrics for grey (GM) and white (WM) matter between the controls and patient group showed a significant (*p* < 0.05) difference in supratentorial WM and GM for all evaluated diffusion metrics, as well as for brainstem, corpus callosum, and thalamus. An intra-subject VS/UWS and MCS group comparison showed only kurtosis metrics and fractional anisotropy differences using tract-based spatial statistics, owing mainly to macrostructural differences on most severely lesioned hemispheres. As a result, we demonstrated an ability of DKI metrics to localise and detect changes in both WM and GM and showed their capability in order to distinguish patients with a different level of consciousness.

## 1. Introduction

The term “consciousness” in clinical practice is defined as the presence of two main components: wakefulness and awareness [1]. If wakefulness is present, but signs of awareness are absent, the patient is referred to having vegetative state/unresponsive wakefulness syndrome (VS/UWS). On the contrary, if any evidence of awareness along with wakefulness cycle can be seen, a clinician may diagnose a minimally conscious state (MCS) [2,3]. The mentioned two states are difficult to differentiate unambiguously since clinical evaluation is often subjective and based solely on clinical behavioural scales that can be influenced by sensorimotor impairment, unnoticeable motor activity, pain, apraxia, aphasia, deafness etc. [4]. Even with the use of established diagnostic criteria, the rate of misdiagnosis remains as high as almost 40% [5,6]. Since the correct diagnosis in disorders of consciousness (DOC) is of great importance for a selection of rehabilitation strategy, recovery prognosis and support of patient family, many researchers try to find reliable objective biomarkers to estimate the level of consciousness. 

Multimodal neurophysiological and imaging methods such as electroencephalography (EEG), transcranial magnetic stimulation-EEG (TMS-EEG), functional MRI (fMRI), MR-spectroscopy, and positron emission tomography (PET) are widely used for assessments of brain structure, its function and metabolism in terms of residual consciousness and cognitive function [7,8,9]. For example, PET studies showed that DOC patients exhibited a decrease in brain regional metabolism of up to 40% of the normal value, and this decrease differs in VS/UWS versus MCS patients [1]. 

With the increasing number of similar studies, it had been revealed that there is no a single structural biomarker for consciousness differentiation, but many of them. The stage of DOC is determined by the severity and number of lesions affecting the brain due to diffuse axonal injury, multifocal cortical contusions, laminar necrosis, thalamic damage, and necrosis [10,11]. Moreover, it was suggested that some regions might serve as the main brain “hubs”, i.e., brain areas that are more important for the function of consciousness than the others [12]. For example, changes of the brain metabolism in DOC patients were most prevalent in regions including frontoparietal areas, midline anterior cingulate/mesiofrontal and posterior cingulate/precuneal associative cortices [13,14]. Damage to these brain “hubs” exhibit a strong correlation with resting-state functional MRI data, and significantly lower activity of the default mode network in DOC [7,15]. Thus, within- and between-network functional disconnections regarding large-scale functional networks of the human brain are considered to be a correlate of unconscious states [1,16,17].

Understanding the origin of the microstructural changes of brain tissues underpinning functional disconnections is also crucial for unravelling the mechanisms of DOC [18]. Conventional MRI techniques may detect and estimate the gross pathological processes leading to coma and other DOC states. There were several attempts to evaluate the DOC prognosis using the standard MRI contrasts (T_2_- and T_1_-weighted images) based on lesion distribution, pathology, and amount of lesions [19,20]. However, popular MRI modalities failed to detect microstructural changes in the case of DOC patients with minimal brain lesions. Diffusion-weighted MRI (DW-MRI) techniques are used for localisation and estimation of brain changes typically not visible by the conventional T_1_- or T_2_-weighted MR imaging. Correlation between functional and microstructural changes in various DOC stages was confirmed earlier in the study that utilised direct comparison of PET and DW-MRI [21]. Thus, the diffusion imaging technique might be a useful technique for differentiation of VS/UWS from MCS patients [22,23,24].

The most widely used diffusion approach in clinical neuroimaging is diffusion tensor imaging (DTI). For example, in DOC patients it has been applied for the prediction of long-term neurological outcomes in acute stage [25], for brain microstructure damage in both acute and chronic stages, including long-term prospective observation [26,27]. DTI assumes that water diffusion can be described by a Gaussian probability distribution in biological tissue [28]. However, water molecules in biological tissue experience complex non-Gaussian processes due to the presence of different barriers such as cell membranes, complex distribution of geometrical constraints etc. Considering this limitation, one can use higher orders of the cumulant expansion of the diffusion propagator in order to partially take into account the complexity of the diffusion processes, for example, by applying diffusional kurtosis imaging (DKI) [29]. In DKI, in addition to the diffusion tensor, deviation from the Gaussian hypothesis is taken into account by the kurtosis of the diffusion displacement probability density function. Thereby, DKI provides a more accurate tissue microstructure characterisation [30] and could be sensitive biomarker in in-vivo studies [31,32,33,34,35,36,37]. Moreover, DKI does not demand a presence of high anisotropic environment such as WM and permits the quantification of the microstructural integrity in general, for example, in the case of GM [29,38,39,40,41]. 

In the present study, we aimed to investigate the question of whether the diffusion kurtosis metrics provide enriched information and a more sensitive chronic DOC differentiation criterion compared to the DTI metrics regarding tissue changes following anoxia and severe chronic TBI. At the same time, we would like to assess the feasibility of DKI metrics as a source of biomarkers for microstructure changes in grey matter in DOC patients.

## 2. Materials and Methods

### 2.1. Patients

We recruited 23 patients who fulfilled the criteria of chronic DOC (VS/UWS, MCS) of traumatic (≥12 months after accident) or anoxic (≥6 months after accident) origin. However, 5 patients were excluded from the study due to severe motion artifacts (4 patients) or excessive brain deformation due to decompressive craniectomy with massive superficial siderosis (1 patient). As a result, we used data obtained from 18 patients (see Figure 1 as an example of patient images). 

In Table 1, we summarised demographic and clinical characteristics of the patients. Control group included 18 age-matched healthy volunteers but 4 subjects were excluded due to a lack of image quality (7 males, 7 females; mean age 40.2 ± 12.8 years). In patients, the clinical examination was performed using the standard tool (Coma Recovery Scale—revised (CRS-R)) on the day of scanning. CRS-R comprises 23 hierarchically arranged items associated with brainstem, subcortical and cortical processes [2]. The lowest score on each subscale represents reflexive activity, while the highest item represents cognitively mediated behaviour by addressing to auditory, visual, motor, oromotor, communicative and arousal functions. The CRS-R is established as the most reliable clinical tool for chronic DOC assessment [42]. 

Whenever it was possible, MRI acquisition was performed without patient sedation. However, in order to avoid severe motor artefacts during the scanning in 8 patients (44%), an anaesthesiologist induced light sedation by dexmedetomidine administration via intravenous infusion at a constant rate of 1 μg/kg·h^−1^. During the infusion period, the anaesthesiologist monitored the patient’s cuff blood pressure, the electrocardiogram, and the pulse oximetry. 

The local ethical committee approved the study. Informed consent was obtained from the legal representatives of patients and from healthy volunteers before any study-related procedures.

### 2.2. MRI Measurements

MRI data were acquired using a Siemens MAGNETOM Verio 3T scanner (Siemens Medical Systems, Erlangen, Germany) with a standard 32-channel matrix head coil. The patient and the control groups underwent standard axial T_2_-weighed imaging (TR 4000 ms; TE 118 ms; slice thickness 5.0 mm; in plane resolution 1.5 mm^2^; duration: 2 min 02 s) in order to evaluate an extend of the brain damage for DOC patients and to exclude any pathology in controls with further diffusion MRI examination using the productive spin-echo echo-planar imaging sequence with three *b*-values (0, 1000, and 2500 s/mm^2^) and 64 non-coplanar diffusion directions for each non-zero *b*-value (TE/TR 115/12600 ms; matrix 100 × 100, resolution 2 × 2 × 2 mm^3^; number of excitation was equal 1, GRAPPA acceleration factor 2). To improve the signal-to-noise ratio and corrections of the susceptibility-based distortions, we acquired two non-diffusion-weighted images with different phase-encoding directions. In addition, a 3D T_1_-weighted anatomical image (MPRAGE) was acquired (TE/TR 2.47/1900 ms, TI 900 ms, flip angle 9°, matrix 256 × 256, 176 axial slices to cover the whole brain) with an isotropic voxel resolution of 1.0 × 1.0 × 1.0 mm^3^. 

### 2.3. Image Analysis

Diffusion image post-processing prior to estimation of the diffusion scalar metrics was done as follows: first, we performed a noise correction for all diffusion images by assuming the non-central χ^2^ distribution due to employed parallel imaging [43]. Next, the data were corrected for susceptibility and eddy-current induced distortions and subject motions using the topup and eddy utility from the FSL package [44]. A correction of the diffusion directions was performed in accordance with [45]. In order to suppress the Gibbs ringing effect [46] and decrease a possible presence of outliers [31,47] we applied Gaussian smoothing with a kernel of 1.5 × 1.5 × 1.5 mm^3^. All diffusion metrics were estimated using the ExploreDTI package [48] using weighted linear estimator. As a result, we obtained fractional anisotropy (FA), mean, radial, and axial diffusivities (MD, RD, AD) and mean, radial, and axial kurtosis (MK, RK, AK) metrics [30,49]. For convenience, we have to note that the diffusion scalar metrics estimated in a frame of the kurtosis model might differ from the estimations of the conventional DTI modality [50].

For the region-based analysis, we used the original T_1_-weighted image for segmentation of the grey and white matter applying the FAST utility from the FSL package [51] with further images binarization and creating GM/WM masks for each subject. Then, each mask was visually inspected in ITK-SNAP viewer [52] with manual correction of the binary masks if necessary. The non-diffusion weighted original image of each subject was aligned and interpolated to its T_1_-weighted reference image using affine transformation and cubic spline interpolation with a normalised mutual information as a quality criterion. The co-registration procedure was visually inspected and applied for all diffusion-derived scalar metrics. The interpolation step allows one to improve the statistical reliability of the following analysis [31,33].

The brainstem and thalamus regions of interests (ROI) were manually segmented using ITK-SNAP [52]. The brainstem ROI included area from craniovertebral junction to the anterior-posterior commissural line, thereafter the thalamus ROI was outlined. The corpus callosum (CC) ROI included 3 continuous mid-sagittal slices of the CC, outlined on T_1_-weighted images. An outline of our post-processing approach is summarised in Figure 2.

### 2.4. Statistical Analysis

We performed statistical analysis based on a group comparison between healthy subjects and three patient subgroups, namely, all patients, MSC patients, and VS/UWS patients, using tract-based spatial statistics (TBSS) [53]. The most representative subject’s structural image was used and aligned to the Montreal Neurological Institute (MNI) 152 standard space using non-linear co-registration utility from FSL. A mean FA image was built up from FA maps of all subjects and thinned in order to produce a mean FA skeleton. We used an FA threshold equal to 0.2 for the FA skeleton to avoid voxels that could be treated as grey matter or as cerebrospinal fluid. Next, all FA images were projected onto the skeleton. At the end, all data were prepared for the voxel-wise statistical analysis. The statistical analysis was performed using permutation-based, voxel-wise non-parametric testing, implemented via the “randomise”-function with 5000 permutations in FSL package. In TBSS analysis we used threshold-free cluster enhancement approach. Resulting p-values were corrected for multiple comparisons and thresholded at *p* < 0.05. Non-FA images were prepared in a similar way for the statistical tests in according with FSL guidance. 

Subsequently, we performed a statistical analysis of the ROI derived diffusion metrics using in-house Matlab scripts (TheMathWorks, Natick, MA, USA) based on the non-parametric Wilcoxon rank sum test. 

## 3. Results

### 3.1. Patients versus Control Group

The diffusion scalar metrics comparison was performed for supratentorial grey and white matter, and separately for the thalamus, corpus callosum and brainstem in control and patient groups (Figure 3, Table 2). In supratentorial WM, we found a significant difference (*p* < 0.05) in the DTI metrics (FA, MD, AD, RD) between control and patient groups, as well as in the DKI metrics (MK, AK, RK). The largest effect was detected for FA values in the case of WM (Cohen’s *d* = 3.28). This result is clearly confirmed by a TBSS analysis (see Figure 4). Due to remarkable brain damage, FA as well as MK, AK, RK in patients was significantly lower, with higher MD, AD, and RD comparing to healthy volunteers. As for GM, the same changes pattern was evident, but without any significant differences for FA (see Table 2). When analysing the thalamus separately, the most reliable difference occurred in MD, AD and RD maps (*p* < 0.001), with higher values in patient group. Absence of significant FA inter-group difference in the thalamus could be influenced by manual segmentation in the patient group since the exact ROI borders are difficult to outline due to severe brain architecture damages in most patients. Then we repeated the analysis taking a smaller thalamus ROI as 3 continuous circle ROIs (diameter 8 mm) in the middle part of both thalamus sides on the level of intrathalamical adhesion. We obtained the same result as for the whole thalamus analysis, with no significant difference in the FA metric. The brainstem analysis revealed significant differences in all metrics, except for RK, with the FA showing the highest values in control group comparing to patients (Cohen’s *d* = 4.66). There were significant differences in all diffusion metrics in corpus callosum, which is the most coherent WM tract, with remarkably higher difference for the DKI metrics comparing to the DTI metrics.

### 3.2. Anoxia versus Trauma Patient Groups

The DOC intra-group comparison based on anoxic versus trauma brain lesion aetiology showed a significant difference only for AK in GM, with anoxic AK values being higher than the traumatic ones (0.55 versus 0.46, Cohen’s *d* = 1.13) (see Figure 5a). The TBSS analysis did not reveal any significant differences in the case of Anoxia versus Trauma group comparison for all metrics.

### 3.3. VS/UWS versus MCS Patients Groups

There was no significant difference between the groups when we compared the metrics using ROI-based analysis (see Figure 5b), but we found a clear tract-specific difference in the TBSS analysis (Figure 6). TBSS analysis showed that DOC intra-group comparison possesses a significant difference (*p* < 0.05) in FA and kurtosis metrics in all types of fibres projection, commissural, associative (Figure 6a), predominantly in the left hemisphere. An increased significance level (*p* < 0.001) reduced a number of useful diffusion metrics and only the kurtosis metrics (MK and AK) localised the difference in the left hemisphere (Figure 6b).

## 4. Discussion

To the best of our knowledge, the present study is a first attempt to apply the DKI approach in order to investigate non-Gaussian diffusion patterns in the brain and associated tissue degradation in chronic disorders of consciousness. We performed a group analysis based on “patients versus controls”, “traumatic versus anoxic patients”, and “VS/UWS versus MCS patients” comparisons. Firstly, we assessed microstructure changes using diffusion kurtosis metrics in DOC patients in contrast to the healthy subjects. In turn, in different subgroups of DOC patients, we investigated a question whether advanced diffusion imaging method works in the frame of severe brain lesions. Therefore, these group comparisons have a methodological implication allowed us to differentiate microstructure alternations for inter- and intra-DOC groups. Moreover, clinically related difficulties in distinguishing different levels of consciousness inspired us to search more reliable biomarkers among diffusion metrics. Taking into account a different aetiology of the patient subgroups, we aimed to assess in vivo variations in neuropathology of traumatic brain injuries and anoxia that might improve a prognostication and to help in treatment planning.

We demonstrated the ability of DKI metrics to detect and localise the changes in both WM and GM and showed their feasibility to distinguish the patients with different forms of chronic DOC. Our results replicated previous studies utilising DTI metrics only [22,23,24,54] and allowed us to assess and differentiate white matter damages in DOC. The results of TBSS analyses suggested that microstructure of almost all WM tracts is dramatically affected in DOC. As a result, the most important structural connectivity disruption patterns were intricate and problematic for predicting the level of consciousness. Along with WM changes, extensive cortex, thalamus and brainstem lesions of different aetiology and their combinations contributed to the consciousness decrease, in accordance with [55]. The use of advanced diffusion approach such as DKI allowed us to evaluate not only WM tract degradation but to perform a GM assessment as well. For example, significant AK differences were obtained for supratentorial GM in DOC subjects of different aetiology. Such diffusion approach seems to be a more robust and reliable method taking into account severe brain damage and deformation of brain structures in chronic DOC patients. It allows us to avoid additional mistakes in fibre tract delineation and its reconstruction, in particular, recalling group analysis methods with brain normalisation algorithms. 

The analysis based on kurtosis as well as diffusion tensor metrics showed here a good impact in distinguishing DOC patients from healthy subjects since not so many diffusion kurtosis research has been performed in clinical settings so far [30,56,57,58,59], including only mild TBI kurtosis studies in acute period where only kurtosis metrics revealed alterations in some brain structures that correlated with cognitive impairment [60,61,62,63]. In our study we found dramatic diffusion metric changes in all regions of interest such as increased MD, AD, RD and decreased kurtosis values in patients comparing to the healthy controls (for example, see Figure 4 and Figure 5). The found tissue degradation with confirmed extensive brain damages in chronic DOC patients affected both WM and GM (see Figure 2 and Figure 3). Notably, all used diffusion metrics represent integrative values and do not reflect the unique microstructure changes and its causes. One can expect the mutually adjusted diffusion changes (for example, MD increase and FA decrease) correlated with more severe brain damage and lower level of consciousness in chronic DOC patients [22,25,26]. We should emphasise that our results are in agreement with the previous findings based on DTI metrics only [22,23,24,55], and provided additional information based on the kurtosis metrics.

FA is one of the important and simple summaries allowing one to probe microstructural integrity of coherent fibres. In the case of DOC patients, the FA shows the highest difference in WM and is associated with severe disruption. Nevertheless, FA does not work for GM (in both cortex and deep grey matter), since there is no high anisotropic tissue environment detectable by DTI at that scale. In opposite, all kurtosis metrics of supratentorial GM were higher in healthy controls as a measure of heterogeneity and complexity of the GM microstructure [36]. At the same time, increased values of diffusivity (MD, AD, and RD) in patient’s GM reflected significant GM degradation. Kurtosis metrics did not reveal any difference in thalamus in contrast to the detected changes of MD, AD and RD values. This discrepancy can be explained by a phenomenological background of the DTI and DKI metrics coming from the cumulant expansion of the diffusion propagator. Thus, an absence of underlying biophysical models based on DTI or DKI allows one to search empiric correlations associated with tissue changes only. 

It is well known that thalamus is severely affected in DOC patients [64]. We found that changes of increased MD, AD and RD values could be a criterion for thalamus degradation and are consistent with the data obtained in [23]. However, severely damaged tissue rarefication due to cells and fibres destruction can be partly compensated by reactive astrogliosis and scar tissue formation with collagen and reticulin fibres laid own, the whole appearing as a glio-mesodermal reaction [65]. These changes might lead to complex microstructure with no significant difference for FA and kurtosis metrics between healthy controls and patients similar to our findings for kurtosis metrics. From this point of view, the reduced MK values can be associated with a loss of cellular structures [60]. In turn, various nuclei of the thalami were altered differently due to cells loss or cells atrophy in association with reactive changes in microglia, macrophages, and astrocytes [66]. Thus, we assume that a ROI-specified analysis of different thalamus regions might be more accurate but tricky because of severe thalamus deformation and size changes. The same explanation can be applied for the inter-group difference for MD, as a possible inverse measure of the membrane density since MD metric could be more influenced by tissue gliosis and reorganisation of membrane barriers. 

Another interesting finding was that AD was increased in all ROIs in DOC patients. Decreased AD metrics in clinical neuroradiology are often considered to be associated with axonal damages and disruption of white matter bundles [67,68], which certainly has place in chronic DOC patients. However, other studies, including simulations and synthetic models, strongly discourage researchers from interpreting changes of the “axial” and “radial” diffusivities on the basis of the underlying tissue structure, unless accompanied by a thorough investigation of their mathematical and geometrical properties in each studied dataset [69]. The direction of the “axial” diffusivity would not be always preserved in pathological tissue and aligned with the underlying real tissue architecture [70]. Thus, explanation could include an additional interpretation such as axonal loss in measured area with interstitial fluid increase, i.e., by cerebrospinal fluid area [41]. As a result, all diffusion eigenvalues are increased with leading increase of AD, RD and MD metrics. 

The highest inter-group difference was observed in corpus callosum where the kurtosis metrics showed more reliable changes (Cohen’s *d* = 7.52 for RK). The radial diffusion metrics (RD, RK) changes in CC experienced prevalence over axial ones. It could be associated with a profound demyelination. Thus, RK may serve as a new method for examining the integrity of the cell membrane and the surrounding myelin sheaths [36,41]. On the other hand, the analysis did not reveal a significant difference in RK for the brainstem between healthy and DOC patients as we found for RD. Morphological changes included not only demyelination and the presence of myelin debris, but axonal damage leading to decrease in the axonal diameter and reactive gliosis. Individual components of this process and the combination of morphological components have different effects on the behaviour of water molecule diffusion. It can lead to intricate diffusion processes in both residual healthy and severe damaged tissue [41]. At the same time, the presence of residual short white matter tracts following other directions than superior-inferior or inferior-superior, and so on, could influence the result and partially explain the absence of RK differences of the brainstem metrics in contrast to CC one.

ROI-based analysis between patients with different aetiology or stage of consciousness did not reveal any significant differences. However, we should mention that possible difficulties in the detection of such difference might originate from the fact that a different combination of brain lesion patterns (in TBI—mainly diffuse axonal injury and brainstem lesions, in anoxia—cortical lesions and selective necrosis of thalamus) may lead to the same clinical presentation of DOC [55]. For example, VS/UWS can be the result of either severe neocortical diffuse damage in asphyxia as well as of severe bilateral thalamic damage due to cardiopulmonary arrest with much of the neocortex spared [11]. Recent DKI studies in traumatic brain injury (TBI) performed either on patients or on animal models, showed that kurtosis metrics correspond to active processes involving reactive astrocytes, in particular, in GM, which cannot be revealed with other MR imaging techniques [61,71]. Diffuse damages of subcortical white matter are common to all chronic DOC patients. In such a way, despite structurally intact cortex the brain function is disabled due to the impairment of connections between different cortical areas via the thalamic nuclei, as DOC is associated with extensive damages to afferent and efferent cerebral connections [64]. In turn, the TBSS analysis did not find any differences based on aetiology. However, it detected WM changes between VS/UWS and MCS groups (see Figure 6). Moreover, the results of TBSS comparison in the case of kurtosis metrics demonstrated a higher sensitivity to structural changes in these subgroups, in particular, with increasing significance level (Figure 6b). The difference is seen mostly in the left hemisphere, and this side-predominance could originate from the fact that the most severe brain lesions were localised in the left hemisphere in 5 VS/UWS patient with the right side being predominantly affected in 2 MCS patients. Thus, a group concerned limitation based on non-uniform brain lesion pattern can affect the TBSS results. However, a found differentiation between MCS and VS/UWS patients offers a great opportunity for non-Gaussian models to become a sensitive biomarker of level of consciousness in chronic DOC in doubtful cases. 

The major limitation of this study is the small sample size. A recruiting of a larger patient cohort would certainly improve a statistical power and diminish spurious or unreliable findings in some anatomical regions. However, we have to note that DOC patients demand a special treatment and operation. It presents some difficulties in recruiting and measurements. We hope to increase the sample size in further ongoing studies. Another problem is an image manipulation and data processing related to the brains with massive lesions and severe deformations [72]. Currently, there is no reliable method of DKI/DTI post-processing similar to reference [73], which allows one to validate and verify a robustness of numerical algorithms in the case of heavy brain deformation/deflection. Thus, it might introduce a larger variability in the results. Moreover, severe brain atrophy with subsequent cerebrospinal fluid enlargement increases bulk motion due to periodic tissue motion caused by cardiac pulsatility, especially in the brainstem. As a result, it could influence the final diffusion metrics [74]. There is also not enough of histological confirmation of the diffusion imaging findings in DOC patients, since the present studies are based on animal models [75] or report a very few number of human observations [76]. As a consequence, subsequent longitudinal studies are strongly required, with DKI/DTI multiple time points measurement, from an acute stage of brain lesion to the chronic DOC with morphology confirmation. 

## 5. Conclusions

In conclusion, we found that the scalar kurtosis metrics represent additional biomarkers, in particular, for detection of micro- and macrostructural brain changes in the case of chronic DOC. AK was found to be sensitive for DOC aetiology differentiation and MK and AK—for VS/UWS versus MCS comparison. The application of non-Gaussian diffusion metrics yield enriched information about status and perspectives of chronic DOC patients. Since the DKI dataset holds DTI metrics as a part of its total diffusion information, it might be very effective to use both types of metrics. Thus, it might help clinicians to improve the diagnostic approaches in DOC and to estimate the prognosis with higher accuracy. 

## Figures and Tables

**Figure 1 brainsci-09-00123-f001:**
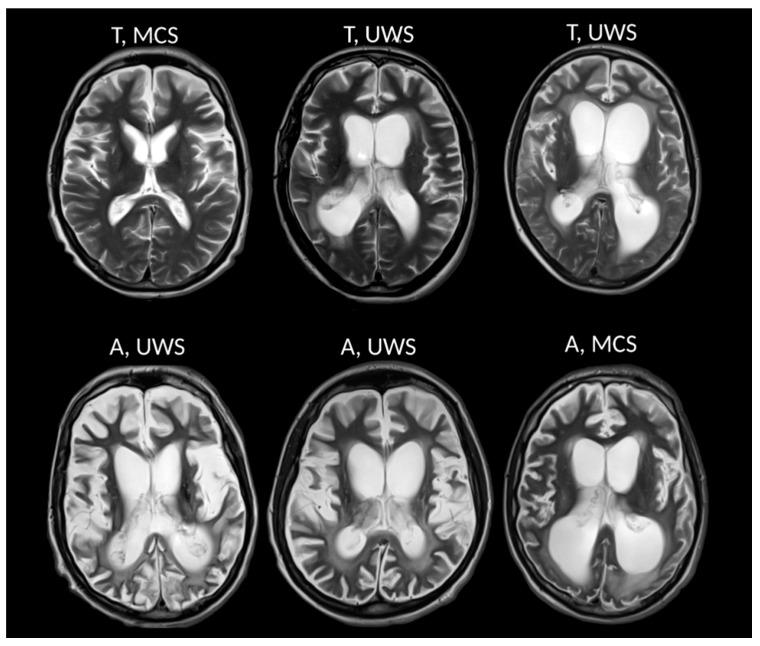
MRI (axial T_2_-weighted image) data present different chronic disorders of consciousness (DOC) patients from the study. Upper row—trauma patients (T), bottom row—anoxic patients (A).

**Figure 2 brainsci-09-00123-f002:**
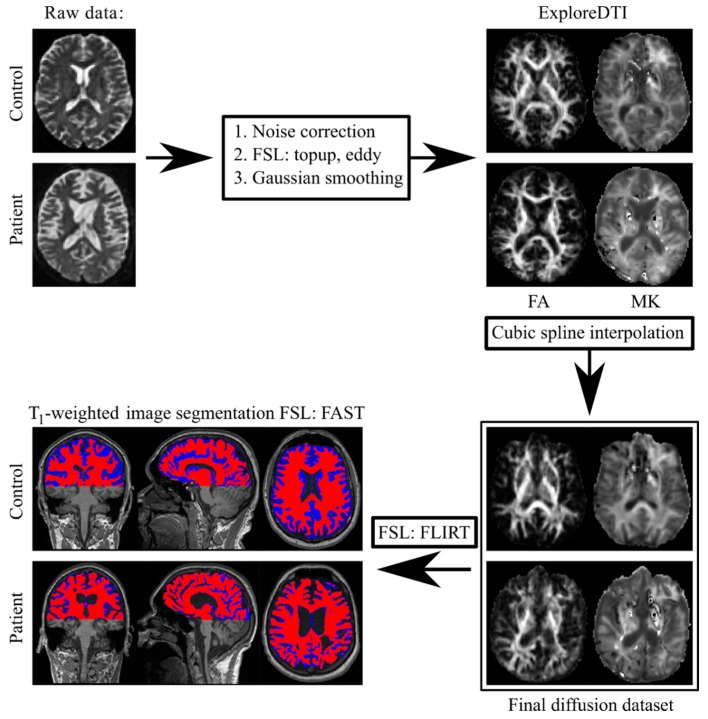
Schematic representation of the diffusion post-processing procedure.

**Figure 3 brainsci-09-00123-f003:**
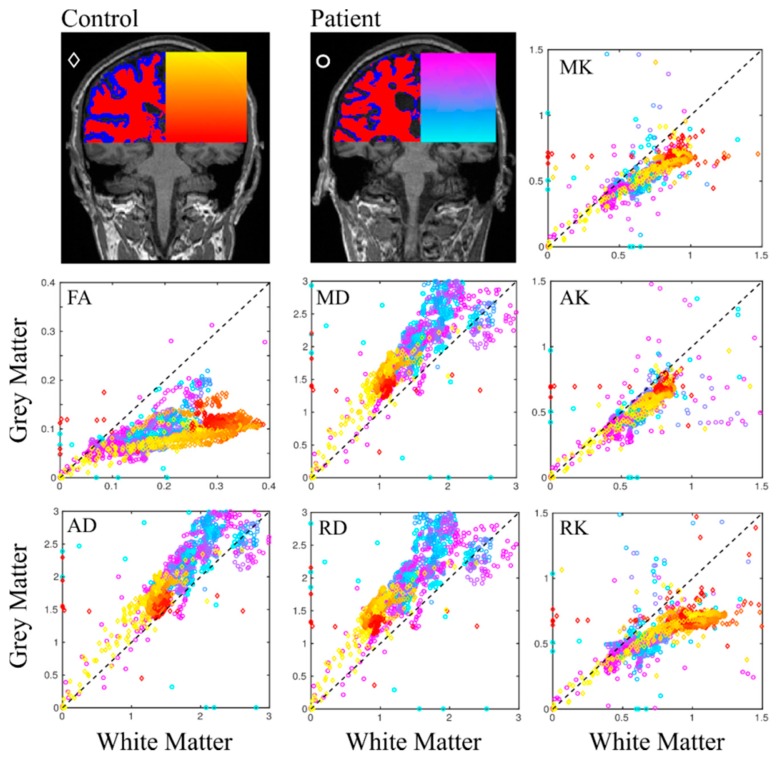
Scatter plots of estimated diffusion metrics for grey/white matter between control/patient groups. Points on the scatter plot are encoded by the colour in respect to the axial slice position and control/patient groups. All diffusion metrics were aligned to T1-weighted individual images. Diffusion units for mean, axial, and radial diffusivities (MD, AD, and RD) are in (µm^2^/ms). The dashed line presents the linear dependence.

**Figure 4 brainsci-09-00123-f004:**
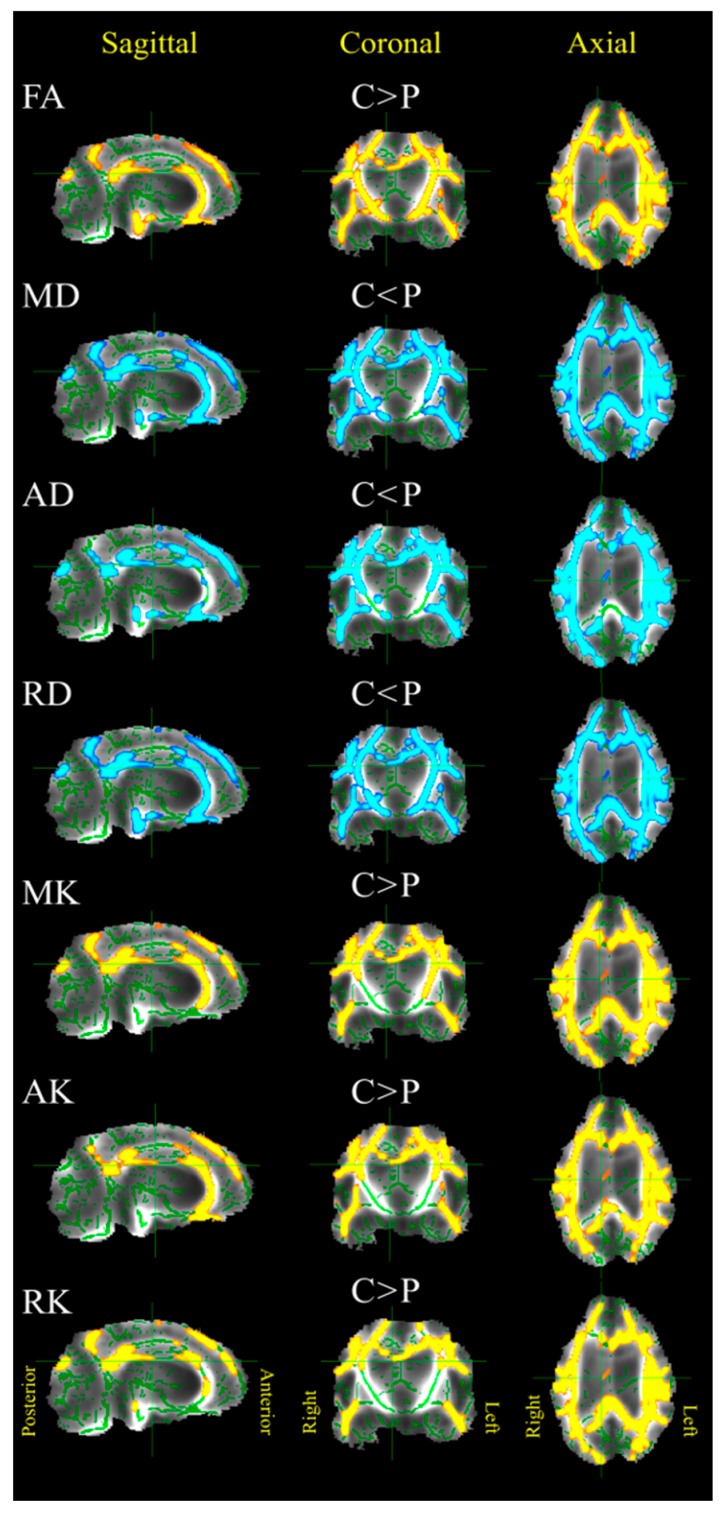
Tract-based spatial statistics (TBSS) results demonstrating locations of metric changes in patients (P) comparing to healthy volunteers (C) (*p* < 0.05). The analysed white matter skeleton is shown in green with the colour overlays of significant differences. Yellow/red colour scheme represents the areas where the diffusion metrics of the patients were significantly lower than in the healthy control group; blue/light blue colour scheme is the areas where the diffusion metrics of the patients were significantly higher, respectively.

**Figure 5 brainsci-09-00123-f005:**
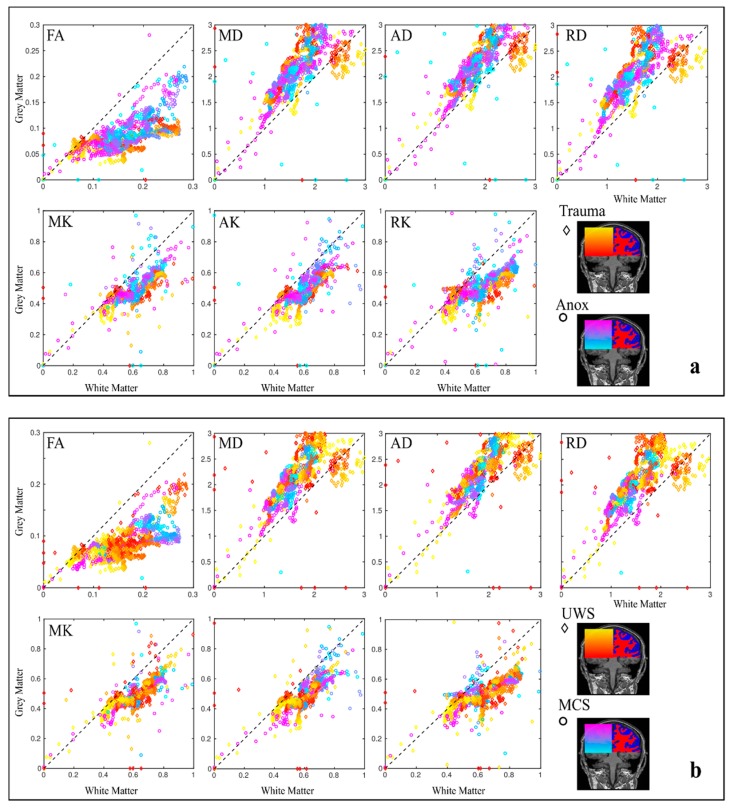
Scatter plots of estimated diffusion metrics for grey/white matter between Trauma/Anox groups (**a**) and unresponsive wakefulness syndrome/ minimally conscious state (UWS/MCS) (**b**). Points on scatter plot are encoded by the colour in respect to the axial slice position and control/patient groups. All diffusion metrics were aligned to T1-weighted individual images. Diffusion units for MD, AD, and RD are in (µm^2^/ms). The dashed line presents the linear dependence.

**Figure 6 brainsci-09-00123-f006:**
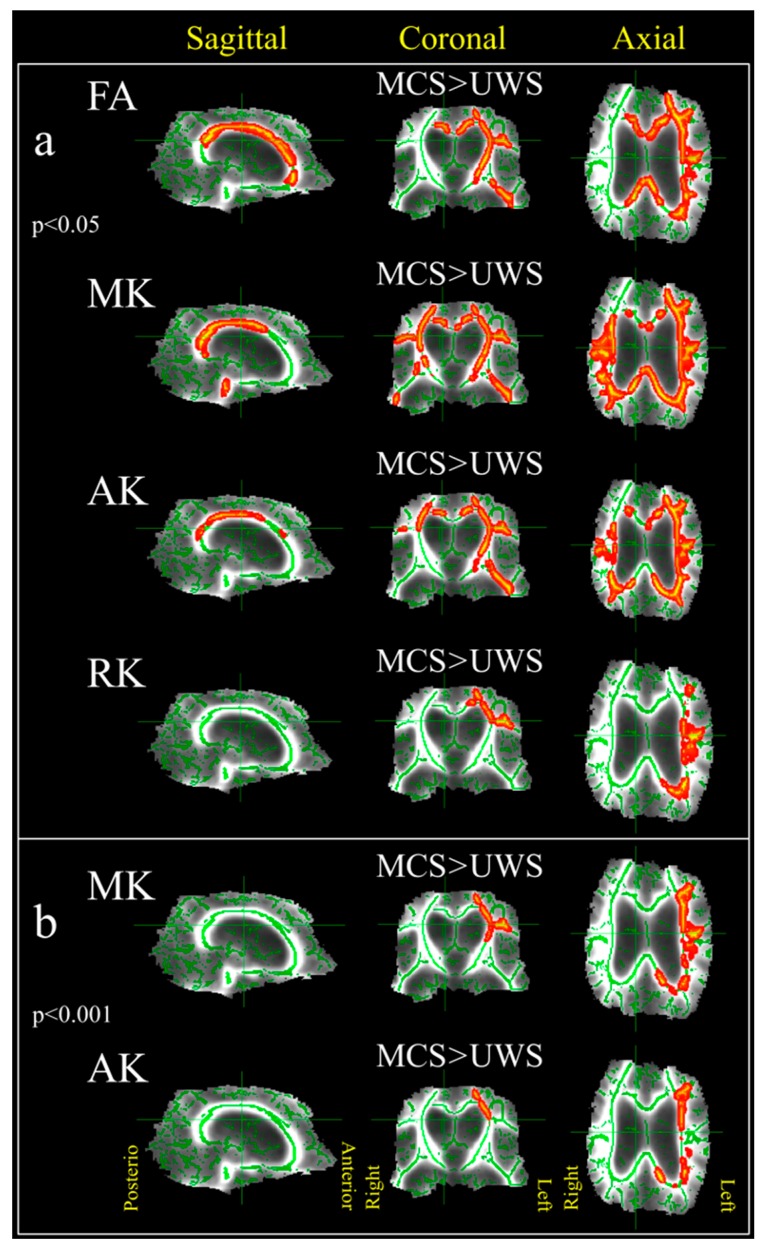
TBSS results demonstrating locations of diffusion metric changes in vegetative state/unresponsive wakefulness syndrome (VS/UWS) versus MCS patients. The analysed white matter skeleton is shown in green with the colour overlay of significant differences. Yellow/red colour scheme represents the areas where the diffusion metrics of MCS patients were significantly higher than VS/UWS. (**a**) the significance level, *p* < 0.05, (**b**) significance level, *p* < 0.001.

**Table 1 brainsci-09-00123-t001:** Demographic and clinical data of patients.

ID	Sex	Age	MPI	CRS-R	Aetiology	Lesion Site	Diagnosis
1	M	20	15	21	T	B	MCS
2	M	48	13	7	A	L > R	VS/UWS
3	F	47	26	5	A	L > R	VS/UWS
4	F	31	5	18	A	L > R	MCS
5	M	23	5	6	A	B	VS/UWS
6	M	26	5	12	T	B	MCS
7	F	24	22	7	T	B	VS/UWS
8	M	23	12	12	T	L > R	MCS
9	M	47	9	12	A	B	MCS
10	M	22	12	5	T	B	VS/UWS
11	F	25	12	7	T	B	VS/UWS
12	M	67	7	6	A	B	VS/UWS
13	M	47	30	6	A	B	VS/UWS
15	F	43	7	13	A	R > L	MCS
15	M	51	16	5	A	B	VS/UWS
16	F	24	73	4	A	B	VS/UWS
17	M	33	10	12	A	R > L	MCS
18	M	42	3	6	A	L > R	VS/UWS

Abbreviations: MPI, months post-ictus; A, anoxic; T, traumatic; M, male; F, female; MCS, minimally conscious state, VS/UWS—vegetative state/unresponsive wakefulness syndrome. Lesion site—predominant lesion hemisphere site; B, both hemispheres, L, predominant left hemisphere, R, predominant right hemisphere.

**Table 2 brainsci-09-00123-t002:** Control (C) versus patients (P) groups comparison of mean/standard deviation diffusion scalar metrics.

	FA	MD (µm^2^/ms)	AD (µm^2^/ms)	RD (µm^2^/ms)	MK	AK	RK
GM		C < P	C < P	C < P	C > P	C > P	C > P
Control		1.5/0.10	1.6/0.11	1.4/0.10	0.66/0.05	0.63/0.03	0.66/0.04
Patient		2.1/0.45	2.3/0.46	2.0/0.45	0.58/0.25	0.52/0.09	0.59/0.23
Cohen’s *d*		1.97	2.05	1.91	−0.4	−1.6	−0.39
WM	C > P	C < P	C < P	C < P	C > P	C > P	C > P
Control	0.28/0.01	1.0/0.08	1.3/0.08	0.9/0.09	0.87/0.05	0.76/0.04	0.70/0.07
Patient	0.17/0.03	1.6/0.40	1.9/0.38	1.5/0.41	0.68/0.22	0.64/0.14	0.98/0.24
Cohen’s *d*	−3.28	1.99	1.86	2.07	−1.13	−1.19	−1.51
Brainstem	C > P	C < P	C < P	C < P	C > P	C > P	
Control	0.32/0.10	1.4/0.19	1.8/0.23	1.1/0.15	0.85/0.13	0.76/0.12	
Patient	0.22/0.04	1.9/0.25	2.3/0.31	1.7/0.26	0.75/0.09	0.68/0.06	
Cohen’s *d*	−4.66	3.03	3.06	3.06	−1.5	−2.05	
Thalamus		C < P	C < P	C < P			
Control		1.2/0.20	1.4/0.23	1.0/0.20			
Patient		1.7/0.33	2.0/0.33	1.5/0.33			
Cohen’s *d*		2.31	2.7	2.17			
CC	C > P	C < P	C < P	C < P	C > P	C > P	C > P
Control	0.44/0.12	1.5/0.17	2.2/0.31	1.1/0.18	0.94/0.13	0.62/0.04	1.28/0.25
Patient	0.20/0.08	2.2/0.42	2.6/0.36	2.0/0.46	0.58/0.12	0.52/0.05	0.60/0.14
Cohen’s *d*	−4.46	2.21	1.42	2.5	−4.17	−2.47	−7.52

Diffusion metrics demonstrated the significant difference (*p* < 0.025) using Wilcoxon rank test are marked as C > (<) P. Wilcoxon one-sided test for the Patient-Control group is used; results are presented for a significant difference (*p* < 0.025). In Table 2 the mean values over the regions are presented. White matter (WM), grey matter (GM), fractional anisotropy (FA), mean diffusivity (MD), axial diffusivity (AD), radial diffusivity (RD), mean kurtosis (MK), axial kurtosis (AK), and radial kurtosis (RK).

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
