# Peer review of "Feasibility of Non-Gaussian Diffusion Metrics in Chronic Disorders of Consciousness"

_brainsci, 2019, doi:10.3390/brainsci9050123_

Round 1
Reviewer 1 Report
The authors investigated the feasibility of non-Gaussian diffusion approach in chronic disorders of consciousness (DOC) which demonstrated diffusion kurtosis imaging (DKI) metrics to localize and detect changes in both WM and GM in different level of consciousness.
Minor:
Line 29 – replace track-based spatial statistics with TRACT-based spatial statistics
Line 162 – replace‘a FA’with ‘AN FA’
Line 198 – ‘occured’to‘occurred’
Line 216 – using one-sided
Line 217 - In WHICH Table?
Line 340 - an extensive damages?
Author Response
We thank the Reviewer 1 that (s)he found our manuscript to be interesting. Please, find our answers for Reviewer’s comments listed below in red
Line 29 – replace track-based spatial statistics with TRACT-based spatial statistics
The text is corrected (line 32 in the corrected manuscript version).
Line 162 – replace‘a FA’with ‘AN FA’
The text is corrected (line 209 in the corrected manuscript version).
Line 198 – ‘occured’to‘occurred’
The text is corrected (line 252 in the corrected manuscript version).
Line 216 – using one-sided
The text is corrected (line 275 in the corrected manuscript version).
Line 217 - In WHICH Table?
The text is corrected (line 276 in the corrected manuscript version) – in Table 2.
Line 340 - an extensive damages?
The text is corrected (line 342 in the corrected manuscript version) – we replaced “massive” to “extensive” as suggested.

Reviewer 2 Report
After carefully review, I found some issues to address
In this study, the authors attempted to apply diffusion-gradient imaging techniques to investigate tissue degradation associated with the non-Gaussian diffusion pattern of the brain in chronic consciousness disorders (DOC). The authors demonstrated the potential to distinguish patients with other forms of consciousness impairment. However, there are some issues that shoud be clearly addressed by the authors. Please refer to this reviewer’s comments.
1. It should be clearly provided the theoretical background in determining the prognosis of chronic disorders of consciousness by MRI-based FA, MD, AD, RD, MK, AK, and RK measurement.
2. In the methodology, the parts of the statistical method should be separately explained.
3. In Figures 3 and 5, the standard color of the patient's diffusion metric does not match with the representative image (Comparing contol (Red Yellow) and patients (Pink Green)?).
4. In each result for comparison between control and patients, it should be described how the authors can identify where statistical significance exists. In particular, for tables 2 and 4, the standard deviation values must be provided including MEAN values.
5. In Figure 3, it seems that the FA, MD, AD, and RD show a significant difference in the patients as compared to the control. However, in table 2, it was shown that the value of FA decreased in the patient image. In order to eliminate such errors in results analysis, it is recommended to display the results in a bar graph with statistical analysis.
6. Clinically, the effect following increasing or decreasing FA, MD, AD, RD, MK, AK, and RK on the diagnosis or prognosis of DOC should be clearly discussed.
7. The intensity of FA, MD, AD, RD, MK, AK and RK are differently investigated when compared between 1. patient and control, 2. trauma and anox, 3. UWS and MCS. The author should provide a logical analysis to be able to use the differences in the values of each of the other comparisons for clinical prognosis and diagnosis.
Author Response
We thank the Reviewer 2 that (s)he found our manuscript to be interesting. Please find our answers for Reviewer’s comments listed below in red
1. It should be clearly provided the theoretical background in determining the prognosis of chronic disorders of consciousness by MRI-based FA, MD, AD, RD, MK, AK, and RK measurement.
We reviewed theoretical background in prognosis of chronic disorders of consciousness based on Gaussian diffusion MRI metrics, such as MD, FA, AD, RD, that is also clearly stated in the articles 21, 25-27:
“Correlation between functional and microstructural changes in various DOC stages was confirmed earlier in the study that utilised direct comparison of PET and DW-MRI [21]. Thus, the diffusion imaging technique might be a useful technique for differentiation of VS/UWS from MCS patients [22-24].
The most widely used diffusion approach in clinical neuroimaging is diffusion tensor imaging (DTI). For example, in DOC patients it has been applied for the prediction of long-term neurological outcomes in acute stage [25], for brain microstructure damage in both acute and chronic stages, including long-term prospective observation [26,27].” (lines 89-98).
As for kurtosis metrics, no studies were performed in DOC patients so far, but the results in other fields (for instance, stroke, tumour or multiple sclerosis [31-37]) – demonstrated that DKI possesses a great impact for clinical applications. We applied DKI in our study with respect to the theoretical background mentioned in lines 98-111 and in the discussion section.
2. In the methodology, the parts of the statistical method should be separately explained.
We added subsection devoted to the statistical analysis (see lines 185-237, with section “2.4. Statistical analysis”)
3. In Figures 3 and 5, the standard color of the patient's diffusion metric does not match with the representative image (Comparing contol (Red Yellow) and patients (Pink Green)?).
We are not sure that correctly understood the Reviewer comments. The colour in Figures 3 and 5 encodes the diffusion metrics in accordance with their slice position. For instance, in Figure 3 the red-yellow colour encodes diffusion metrics for healthy participants. In turn, the blue-pink colour performs a visual arrangement for the patient's group. This type of visualisation allows us to emphasise a distribution of diffusion metrics depending on the anatomy, i.e. covering regions from brainstem and deep grey matter up to cerebral cortex. In Figure 5 we used the same scheme for a) trauma vs anox groups and b) UWS vs MCS groups, respectively. As one can see in the case 5a it is difficult to distinguish the distributions of diffusion metrics between the groups. While in Figure 5b we are able to identify the visual difference between the groups (see, for example, FA, MD, and RD).
4. In each result for comparison between control and patients, it should be described how the authors can identify where statistical significance exists. In particular, for tables 2 and 4, the standard deviation values must be provided including MEAN values.
We provided mean/std values in Table 2 (there is no Table 4 in the manuscript), and described how significant difference was identified.
5. In Figure 3, it seems that the FA, MD, AD, and RD show a significant difference in the patients as compared to the control. However, in table 2, it was shown that the value of FA decreased in the patient image. In order to eliminate such errors in results analysis, it is recommended to display the results in a bar graph with statistical analysis.
Unfortunately, we cannot identify suggested by the Reviewer difference in the diffusion metrics. In Figure 3 (FA) for GM it is difficult to detect the difference between controls and patients (y-axis). However, in the case of WM one can expect that FA values for the controls are higher, what is proposed in Table 2. In the case of MD, AD, and RD one can see that metric distributions of control group are closer to (0,0) point in contrast to the patient group, what corresponds to similar results in Table (C < P).
6. Clinically, the effect following increasing or decreasing FA, MD, AD, RD, MK, AK, and RK on the diagnosis or prognosis of DOC should be clearly discussed.
We added a speculation related to DTI/DKI metric changes in the Discussion section (lines 344-348), and the further Discussion text explains each metric and its clinical correlation in detail.
7. The intensity of FA, MD, AD, RD, MK, AK and RK are differently investigated when compared between 1. patient and control, 2. trauma and anox, 3. UWS and MCS. The author should provide a logical analysis to be able to use the differences in the values of each of the other comparisons for clinical prognosis and diagnosis.
We are not sure that understood correctly the Reiviewer comment. Analyses for all groups 1. patients vs controls; 2. trauma vs anox; and 3. UWS vs MCS were performed uniformly. We applied TBSS comparison between groups and statistical analysis for manually chosen region of interests. In accordance with clinical use of our findings we added a speculation related to the diffusion metric changes in the Discussion section (see answer above).

Round 2
Reviewer 2 Report
Most concerns have been well addressed. This reviewer would like to ask again about the previous comment #7. The authors have investigated with comparing 3 groups. Group 1 was “patient vs control”, group 2 was “trauma vs anox”, and group 3 was “UWS vs MCS”.
1. It is suggested to describe that the rationale why the authors proposed the experimental design divided into the three groups.
2. It is suggested to describe that the clinical implications that can be inferred from the differences between the results from each comparison.
Author Response
Dear Hannah Mao!
We are grateful for the opportunity to submit a revised version of our manuscript for your consideration. We have added our responses to the Reviewer 2 below and marked the relevant text edits in the manuscript in yellow after using Review mode (Track-mode) of Microsoft Word.
Reviewer 2.
We thank the Reviewer that (s)he found our manuscript to be interesting. Please find our answers for your comments listed below in red
1. Most concerns have been well addressed. This reviewer would like to ask again about the previous comment #7. The authors have investigated with comparing 3 groups. Group 1 was “patient vs control”, group 2 was “trauma vs anox”, and group 3 was “UWS vs MCS”.
1.1. It is suggested to describe that the rationale why the authors proposed the experimental design divided into the three groups.
1.2. It is suggested to describe that the clinical implications that can be inferred from the differences between the results from each comparison.
We added the explanation in the discussion and conclusion:
Lines 303 – 316:
“We performed a group analysis based on “patients vs controls”, “traumatic vs anoxic patients”, and “VS/UWS vs MCS patients” comparisons. Firstly, we assessed microstructure changes using diffusion kurtosis metrics in DOC patients in contrast to the healthy subjects. In turn, in different subgroups of DOC patients, we investigated a question whether advanced diffusion imaging method works in the frame of severe brain lesions. Therefore, these group comparisons have a methodological implication allowed us to differentiate microstructure alternations for inter- and intra- DOC groups. Moreover, clinically related difficulties in distinguishing different levels of consciousness inspired us to search more reliable biomarkers among diffusion metrics. Taking into account a different aetiology of the patient subgroups, we aimed to assess in vivo variations in neuropathology of traumatic brain injuries and anoxia that might improve a prognostication and to help in treatment planning.”
Lines 328-330:
“For example, significant AK differences were obtained for supratentorial GM in DOC subjects of different aetiology.”
Lines 439-441:
“However, a found differentiation between MCS and VS/UWS patients offers a great opportunity for non-Gaussian models to become a sensitive biomarker of level of consciousness in chronic DOC in doubtful cases.”
Lines 464-465:
“AK was found to be sensitive for DOC aetiology differentiation and MK and AK – for VS/UWS versus MCS comparison. ”
